# Analysis of the earliest complete mtDNA genome of a Caribbean colonial horse (*Equus caballus*) from 16th-century Haiti

**Nicolas Delsol**[1]*, **Brian J. Stucky**[1,2], **Jessica A. Oswald**[1,3], **Elizabeth J. Reitz**[4], **Kitty F. Emery**[1], **Robert Guralnick**[1]

**1** Florida Museum of Natural History, University of Florida, Gainesville, Florida, United States of America, **2** Agricultural Research Service, U.S. Department of Agriculture, Beltsville, MD, United States of America, **3** Biology Department, University of Nevada, Reno, Reno, NV, United States of America, **4** Georgia Museum of Natural History, University of Georgia, Athens, Georgia, United States of America

* ndelsol@ufl.edu

**Data Availability Statement:** The raw reads have been deposited on the NCBI SRA repository (https://www.ncbi.nlm.nih.gov/sra, BioProject# PRJNA817535, SRA# SRR18395639). The

## Abstract

Unlike other European domesticates introduced in the Americas after the European invasion, equids (Equidae) were previously in the Western Hemisphere but were extinct by the late Holocene era. The return of equids to the Americas through the introduction of the domestic horse (*Equus caballus*) is documented in the historical literature but is not explored fully either archaeologically or genetically. Historical documents suggest that the first domestic horses were brought from the Iberian Peninsula to the Caribbean in the late 15th century CE, but archaeological remains of these early introductions are rare. This paper presents the mitochondrial genome of a late 16th century horse from the Spanish colonial site of Puerto Real (northern Haiti). It represents the earliest complete mitogenome of a post-Columbian domestic horse in the Western Hemisphere offering a unique opportunity to clarify the phylogeographic history of this species in the Americas. Our data supports the hypothesis of an Iberian origin for this early translocated individual and clarifies its phylogenetic relationship with modern breeds in the Americas.

## Introduction

Despite considerable speculation, the history of the domestic horse in North America remains unclear. We reconstructed the complete mitochondrial genome of one of the earliest specimens of domestic horse (*Equus caballus*) in the Americas in order to better understand the origins and trajectory of the first horses introduced by the Europeans after 1500. The specimen is from a 16th-century archaeological deposit at the site of Puerto Real, Haiti. Puerto Real was one of the first towns founded by the Spanish in the Americas and served as an important port. Horses, cows (*Bos taurus*), and other domestic animals of Eurasian origin were vital to the town [1, 2] and according to early colonial chronicles, all of these animals were brought to Hispaniola from the Iberian Peninsula via the Canary islands [3]. Other than the documentary evidence, little is known about source populations and the spread of domestic horse

annotated mitochondrial consensus sequence has been deposited on Genbank (https://www.ncbi.nlm.nih.gov/genbank/) under the reference # ON168403. Both dataset will be publicly released upon publication of the article.

**Funding:** KFE and ND received support from NSF DDRIG grant (#1930628) for this research (https://beta.nsf.gov/funding/opportunities/archaeology-and-archaeometry-0). ND's initial research was also funded by a Fulbright scholarship. The field research in Haiti was financed through contributions of the Organization of American States, the National Endowment for the Humanities, Dr. William Goya, the Wentworth Foundation, the University of Florida Division of Sponsored Research, and the Florida Museum of Natural History. The funders had no role in study design, data collection and analysis, decision to publish, or preparation of the manuscript."

**Competing interests:** The authors have declared that no competing interests exist.

throughout the Americas. Here we use mitogenomics to place our sample in a phylogeny composed of global horse samples to understand the phylogeographic history of Western Hemisphere horses. Although sampling is limited to a single specimen, it is the oldest horse mitochondrial genome of the post-Columbian era sequenced to date, and of significance for understanding of the introduction of domestic horses in the Western Hemisphere.

## A brief history of horses in the Americas

The *Equus* genus first appeared on the North American continent during the Pliocene era [4] and spread to and across Eurasia beginning around 2.5 million years ago [5]. All equids disappeared from the Western Hemisphere during the megafauna extinction event at the end of the Pleistocene and the last glacial period [6]. It was not until the late 15th-century and the arrival of European explorers in the Caribbean that equids returned to the Americas, this time as *Equus caballus*, the domestic horse.

The introduction of the domestic horse began with the second voyage of Columbus. In 1493, a royal ordinance from the Castilian monarchs ordered Fernando de Zafra to hire Andalusian horsemen to accompany Columbus on his second exploration [7]. Oviedo y Valdés [3] reports that, after Columbus boarded these horses on ships in the Canary Islands, they arrived on Hispaniola, in the town of La Isabela (Dominican Republic), on November 28, 1493.

Over the next decade, the horse population on Hispaniola grew rapidly from a few dozen individuals to large herds, thanks to favorable environmental conditions and continued importation of animals from the Iberian Peninsula. Eventually, the governor of the West Indies, Nicolás de Ovando, decreed there was no need to import more horses because the population size was sufficient on the island [7]. In this same decade, horses were taken to other Caribbean islands as Spanish interests expanded. By the 1520s horses had reached the Mesoamerican mainland and by 1538, the northern coast of the Gulf of Mexico in what is now Florida (USA) [8, 9]. A permanent Spanish presence was established on the North American coast in 1565. One of the most iconic episodes of equid introduction in the Americas is the arrival, in 1519, of Hernan Cortés' 16 horses on the Mexican Gulf coast, contributing to the military conquest of Mexico. These animals were so essential that Spanish chronicler Bernal Diaz del Castillo gave a detailed description of each animal, with its name and appearance [10].

In later years, horses from other regions in Europe were brought to the Americas. For example, in the 1620s, in the British-sponsored colonies of New England, horses were brought from northern Europe, mostly from England and the Netherlands [11]. These later imports were of diverse breeds, some of them larger and better suited to labor than were the earlier Spanish imports [12]. Horse breeding rapidly became an important activity in some British colonies. By the 18th century, New England horses were a major export to the Caribbean where the sugarcane industry was in critical need of draft animals [13].

Horses were not only pivotal to the European military expeditions in the Americas, they also held a crucial role in the implementation of post-Columbian industries such as ranching in which horses were used to manage the wide-ranging cattle [14–16]. With the rapid growth of cattle herds, many horses became feral. By the 1650s they were widespread from Mexico to the Great Plains of North America, where they were tamed and re-domesticated by Native American communities of the Great Plains [15, 17]. In many regions horses deeply impacted Native cultures by providing greater mobility through an equestrian lifestyle [17, 18].

## Horse lineages and mtDNA

The complete mitochondrial genome of the modern horse, which matrilineal lineages, was sequenced for the first time in 1994 [19]. Other modern horse mitogenome studies have

targeted small portions of the mitochondrion, focusing particularly on clarifying the early phylogenetic history of horse domestication [20–23]. These studies revealed a weak geographic structure among mitochondrial haplogroups, the highest diversity of which are found in Asian horses, and also show that European and Middle Eastern horses lack genetic representation of the most ancestral lineages, suggesting an Asian origin [23]. More recently, ancient DNA analyses of archaeological horse specimens focusing on the whole genome (i.e. including the nuclear genome) of ancient horses confirmed the genetic homeland of modern horses as central Asia, and revealed that modern domestic horses probably originated around 4,200 years ago on the steppes near the Volga and Don rivers, now part of Russia, before spreading across Eurasia, ultimately replacing all pre-existing horse lineages [24].

The mitochondrial data from Modern horses suggest that domestic horses emerged via several distinct domestication events from populations with high maternal haplotypic diversity, and a later mixing of these different haplotypes during the early stages of domestication [23]. This contrasts with the evolutionary history of all the other Near Eastern domestic ungulates which arose from a limited number of animals domesticated in a small number of places around 10–8 ky BP [25, 26]. In contrast, studies of ancient horse lineages from the Y chromosome of modern individuals show almost no diversity of past male lineages, which contrasts with the great diversity of matrilineal lineages [27]. Together, these studies suggest that the domestication of horses involved a few closely related males and a very diverse population of females [27].

In the only study of the population genetics of present-day horses of the Western Hemisphere, an analysis of the control region of the D-loop in modern individuals of Iberian and American horse breeds reveals that Iberian animals were more diverse than any American horse breeds, and that many Western hemisphere breeds present a high frequency of haplotypes that can be traced back to the Iberian Peninsula, indicating that most have Iberian ancestry [9]. It also showed that among the Western Hemisphere breeds, diversity is highest in the Caribbean (based only on the Puerto Rican Paso Fino breed), followed by the South American breeds, and then the North American breeds which have the lowest genetic diversity.

Relatively little is known about the introduction of horses to the Americas from their archaeological remains, in part because early colonial horse remains may be mistakenly identified as either pre-extinction equids or Historic/modern intrusions [28]. Horse remains are rare in most 16th-century vertebrate assemblages of the Americas. For example, they represent only 2.3% of the faunal remains in early colonial deposits at the site of Ek' Balam in Yucatan, Mexico [29], 1.75% of faunal remains from an Indigenous midden the site of El Japón in Mexico City [30], and 0.23% of a Spanish elite midden at the site of Justo Sierra in Mexico City [31]. This relative lack of specimens also could be related to the composition of archaeological faunal deposits and the status of horses in colonial societies. Many archaeological deposits are garbage dumps primarily containing food waste and debris from artisan activities. In colonial Hispanic society, horses were highly valued animals used in a multitude of different contexts, e.g. agriculture, transportation, or warfare, but they were not considered appropriate for human consumption, and are unlikely to be recovered amongst food waste [32]. Ancient DNA studies of archaeological horse remains in the Americas are equally rare. While several studies analyzed the extinct lineages of pre-Holocene horses [6, 33] only one published study contains genomic data (no mtDNA) of an historical horse specimen [28]. Thus, the Puerto Real specimen and its ancient DNA presented here is critical to understanding the history of domestic horses in the Americas.

Our study focuses on a single archaeological horse specimen recovered from excavations at the post-Columbian site of Puerto Real founded by Spain in 1503, and the final resting spot of some of the earliest colonial horses. While limited to a single specimen, this study is of

significance to our understanding of the introduction and phylogeography of domestic horses in the Western Hemisphere as it constitutes the earliest complete equid mitochondrial genome sequenced to date for the post-Columbian era. Our approach focuses on the whole mitochondrial sequence, as earlier studies have demonstrated that the analysis of shorter sections of mtDNA do not provide sufficient detail to allow an assessment of the phylogeny of a species in all its complexity [23]. This paper characterizes this genome by comparing it with published equid sequences, situating it in the horse phylogeographic history. Our study provides a unique opportunity to address the question of the origins of the first colonial horses and their relationships to European breeds of the time, and also reveals genetic affinities between colonial horses from the Caribbean and modern populations from North America. This archaeogenomic evidence discusses the early colonial horse in the broader context of 16th-century Eurasian domesticates in the Americas, and contributes to our understanding of the history of the Spanish colonization, highlighting the long-term presence of horses alongside Spanish explorers and colonists on the Atlantic coast of North America.

## Study area

Puerto Real was an early sixteenth-century town in what is now Haiti (Fig 1) [2, 34]. It was founded in 1503, one of the first 15 towns founded on Hispaniola. Nearby Puerto Plata was officially the last port of call for ships sailing from the Americas to Spain until 1515, when Havana, Cuba, was founded [35]. The site was home to a diverse population of Europeans, Taino Natives, and Africans. The economy of Puerto Real primarily was based on mineral resources (copper) and cattle. Colonists exploited cattle hides, which were more valued than meat, and exported them and other cattle products to Europe [1, 36]. Much of this trade was with Portuguese, Dutch, French, and English ships that called at Spanish ports on the northern coast of Hispaniola [36]. This was a violation of Spanish mercantile policies that gave the Spanish *Casa de Contratación* a monopoly on the American trade. Repeated efforts to counteract pirates and corsairs, limit the extensive illegal trade with foreign vessels, and exert more control over commerce on Hispaniola failed. Accordingly, the Crown ordered Puerto Real and other northern ports to be abandoned in 1578 and Puerto Real was destroyed by Spanish officials in 1579, though a few people remained in temporary housing at the site as squatters. The area was abandoned by the remaining Spanish, African, and Native residents in 1605, when the colonial authorities ordered the entire population of the western part of Hispaniola to relocate. After that date, most people on the northern coast were buccaneers, sailors licensed by colonial powers to prey on Spanish colonies, who hunted the herds of feral cattle left behind when Spain abandoned the coast [31].

Excavations at Puerto Real were conducted by Dr. Kathleen Deagan, Curator Emerita of the Florida Museum Historical Archaeology Division, between 1979 and 1990. The 16th-century town was arranged in a 500 m x 400 m rectangle [37]. The excavations revealed several buildings including the church and its cemetery, two elite households and a middle social status residence [2].

## Material

The zooarchaeological collection from Puerto Real represents an invaluable source of data on animal use and human-animal relationship in the colonial Caribbean. The collection was originally identified and analyzed by several researchers including E. Reitz, B. McEwan, and R.A. Marrinan [1, 38]. The Puerto Real faunal materials are curated by the Environmental Archaeology Program of the Florida Museum (the horse specimen is curated in EAP Accession number 0295) and cultural materials are curated by the Historical Archaeology program (cultural

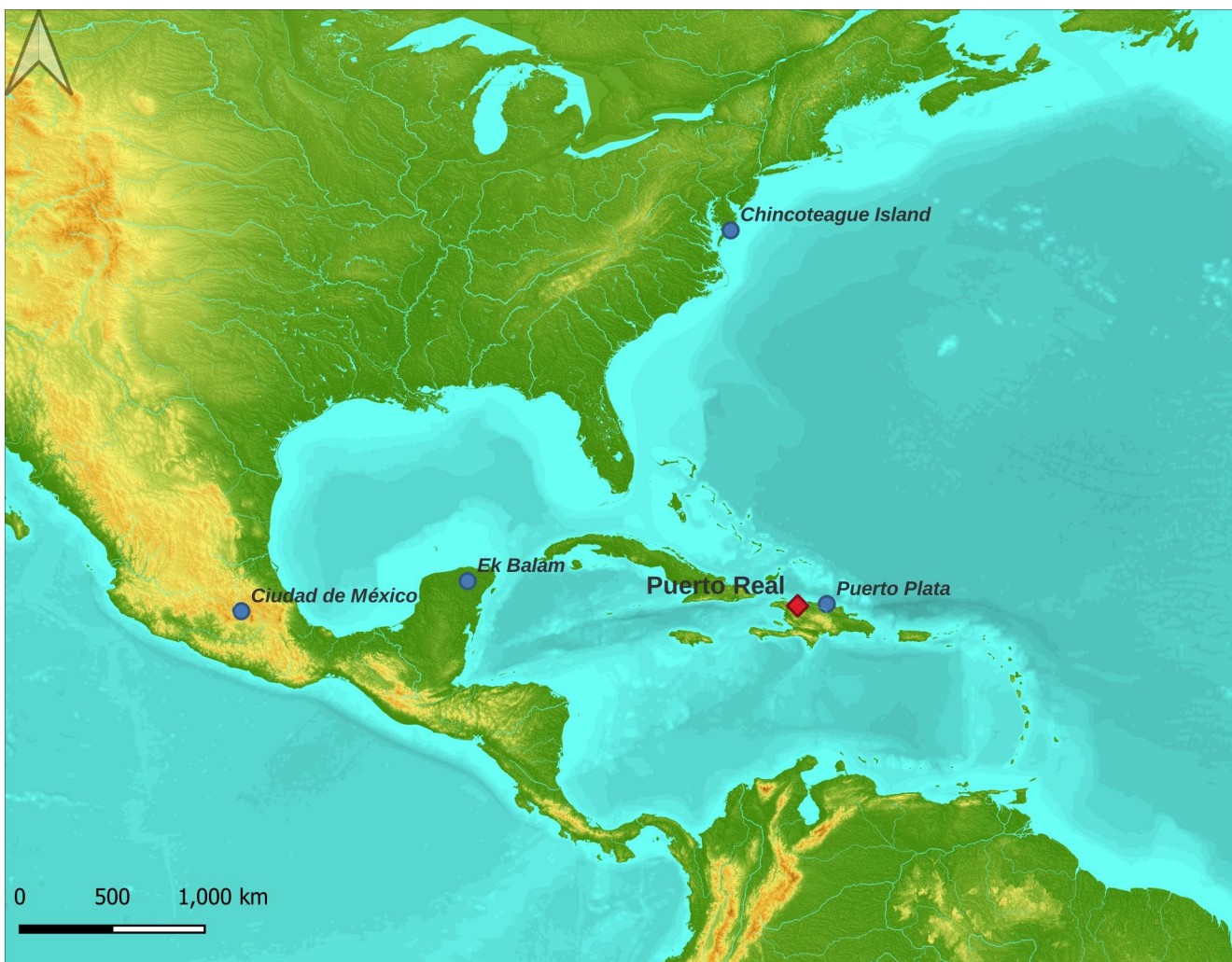

**Fig 1. Location of the site of Puerto Real and other locations mentioned in the study (original map by N. Delsol, cartographic data from USGS EROS http://eros.usgs.gov/, source: NASA/METI/AIST/Japan Spacesystems and U.S./Japan ASTER Science Team).**

materials associated with the horse specimen are curated in ANT 92–002). Of the over 127,000 vertebrate specimens identified from Puerto Real thus far, only eight were attributed to horse when this study began [38]. Horse equipage was recovered from the town, including harness and tack artifacts. C.R. Ewen reports 31 items in this category from Locus 19 [39], McEwan reports three from Locus 33/35 [40], and Locus 39 contained a horseshoe and six harness/tack fragments [38].

The horse specimen used in this study (FLMNH Environmental Archaeology catalog number 02951527, see Fig 2) is a small tooth fragment initially identified as a cow (*Bos taurus*) by the lead author and included in a genetic study of early colonial cattle. Its identification as a horse came about during the genetic analysis itself. The specimen is a lower third molar from the left side. Its occlusal surface presents moderate wear, which implies the individual was an adult horse. The specimen was recovered during Deagan's 1980 fieldwork campaign in an area originally called "Building A" (EAP Accession #0295), later interpreted as the church of the Spanish town [41]. The tooth is from Unit 770N-764E located just north of the church. Stratigraphically, it is from a level deposited during the initial abandonment of the town in 1578 and

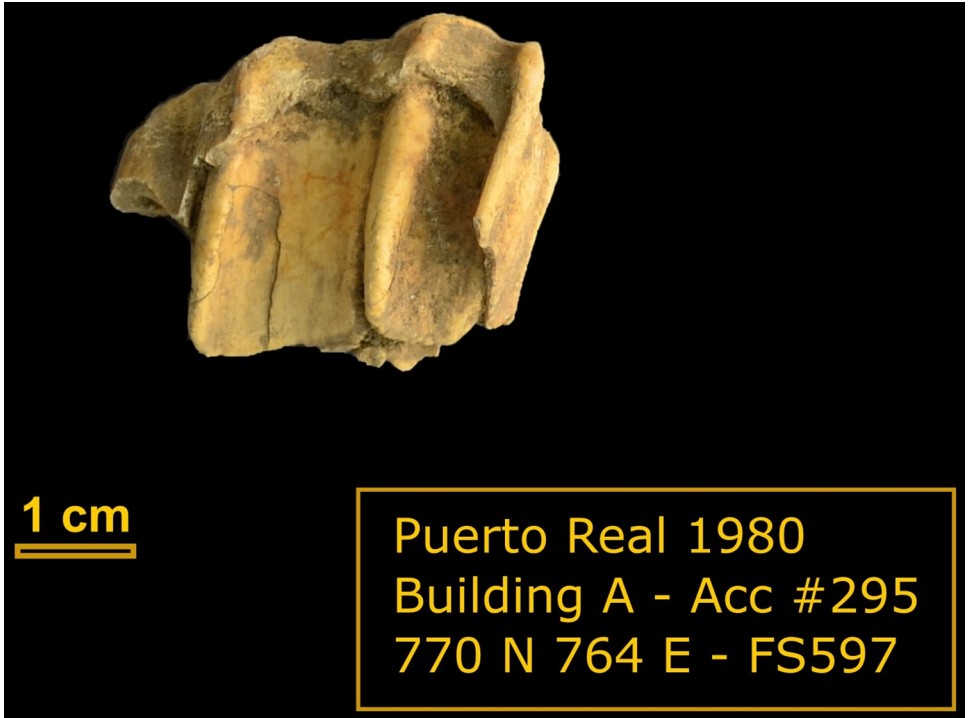

**Fig 2. Picture of the horse specimen R0121-03 (FM-EAP Catalog Number 02951527).**

sealed by a sterile layer above it (i.e. a layer containing no archaeological materials). This archaeological context contains exclusively 16th century materials. Historical and archaeological contexts and material culture permit the excavators to reliably date the horse tooth to the last quarter of the 16th century. The deposit, therefore, represents activities of remnant squatters at the site between 1578 and 1605, after which the town was burned and remaining residents removed. Historical and archaeological contexts and material culture permit the excavators to reliably date the horse tooth to the last quarter of the 16th century.

## Methods

### Data acquisition and permits

All necessary permits were obtained for the described study, which complied with all relevant regulations. Permission for the excavation, import, and curation of the materials were provided by the Haitian government to the Florida Museum in 1980. Permission for use of the specimen in this research (FM-EAP Catalog Number 02951527) was provided following evaluation by the curators and collection managers of the Florida Museum Divisions of Historical Archaeology (Curator Charles Cobb, and Collection Manager Gifford Waters) and Environmental Archaeology (Curator Kitty Emery and Collection Manager Nicole Cannarozzi)in accordance with the Florida Museum Destructive Analysis Policy (https://www.floridamuseum.ufl.edu/wp-content/uploads/sites/32/2017/03/Destructive_Sampling_Policy.pdf)

### DNA extraction and sequencing

All DNA extraction work was conducted in the Florida Museum Ancient DNA laboratory that includes a clean room facility specifically dedicated to the extraction of ancient and historical

DNA samples. We first isolated a fragment of the molar weighing about 500 mg. This sample was then frozen using liquid nitrogen and pulverized with a stainless-steel mortar and pestle. Following Yang and colleagues' protocol [42] as implemented by Oswald et al. [43], the bone powder was combined with an extraction buffer composed of 949 μl of 0.5 M ethylenediamine-tetraacetic acid (EDTA), 25 μl 20 mg/ml proteinase K, 21 μl of 10 mg/ml 1,4-dithiothreitol (DTT), and 5 ul of 10% sodium dodecyl sulfate (SDS). We incubated the sample at 60°C for 24 h, centrigugated it, and then intermittently vortexed and then concentrated it with two Amicon® Ultra-4 Centrifugal Filter Unit and purified using a Qiagen QiaQuick MinElute Kit and eluted in 48 μl of elution buffer according to the manufacturer's directions. A negative control was included alongside each extraction to monitor contamination. DNA extraction of the sample and the negative control were quantified with a Qubit® 2.0 Fluorometer.

After the extraction, we sent the sample to Rapid Genomics (Gainesville, Florida) for library preparation, mtDNA enrichment, and sequencing. Rapid Genomics is a private company dedicated to the processing of genomic samples and sequencing. They followed a strict protocol adapted to the processing of ancient and degraded samples. This protocol was designed in collaboration with the authors. After initial quantification, the libraries were built using Swift Methyl Seq Kits. This kit uses a uracil-tolerant polymerase and performs well with degraded and low-yield samples. Bisulfite conversion was not performed on the horse sample. To retain lower molecular weight fragments and improve libraries with degraded DNA, SPRI bead cleanup ratios were modified as follows: post-extension SPRI ratio 1.8; post-ligation SPRI ratio 1.6; post-PCR SPRI ratio 1.6. Otherwise, the protocol was followed per manufacturers' instructions. For the enrichments, Rapid Genomics designed RNA bait kits using 12,000 probes based on the domestic cow (*Bos taurus*) mitogenome (cattle mitochondrial reference genome, NC_006853.1). Capture reactions were performed on the library using all the library product or up to 500 ng following Rapid Genomics customized workflow. The hybridization was performed at 60°C for 48 hours. After hybridization, enrichment PCRs were performed for 15 cycles. The sample was sequenced on an Illumina MiSeq sequencer (2x150 paired-end sequencing).

## NGS reads processing

The quality of the reads from the sequencing was first assessed using FASTQC [44]. Adapter removal (i5 and i7 Illumina primers) and quality trimming [28] was performed with the program adapterremoval [45]. The forward and reverse reads were then merged into a single read and duplicates were removed in Geneious Prime (https://www.geneious.com/,version 2021.1). The sequence was attributed to a horse after a BLAST search on the reads with the blastn command line application in Geneious using the nt database and the default options (E-value 1e-03). We mapped the reads to the horse mitochondrial reference genome (Genbank NC_001640.1) using the Geneious mapper algorithm set on "Custom sensitivity" with a minimum mapping quality of 30, allowing for only 5% mismatch between reads and not allowing gaps. The Geneious algorithm maps reads to the reference up to 5 times. A contiguous mitochondrial genome sequence was generated by Geneious with a threshold of 75% and quality set on "Highest". The consensus generated with Geneious out of contig pileups represents a mean coverage of 5622.9X across the horse reference mitochondrion. The damage patterns of the mitogenome assembly were assessed using MapDamage [46]. The DNA damage patterns for the mapped mitogenome of the Puerto Real horse are consistent with those of ancient DNA, with elevated C-T transitions occurring on the 5' and 3' ends (S1 Fig). This confirms a lack of contamination from modern DNA. Following analysis, the raw reads were uploaded on the Sequence Read Archive (SRA, accession number PRJNA817535) and the assembled sequence on Genbank (accession number ON168403).

### Phylogeny and haplotype map

The mitochondrial genome of the Puerto Real horse was aligned with the 83 mtDNA sequences of modern horses published by Achilli and colleagues [23] and obtained from GenBank to establish the closest relatives of this individual (S1 Table). This operation was performed using Geneious alignment algorithm with a gap open penalty of 12 and a gap extension penalty of 3. We processed the FASTA alignment file through GBlocks to remove ambiguously aligned sequences [47] resulting in 16,652 bp in the final alignment. To identify the Puerto Real horse sequence and its relationship to known matrilineal haplogroups, we applied a maximum likelihood (ML) model using RAxML [48] with the GTRCAT model of rate heterogeneity (S2 Fig) and 10,000 bootstrap replicates (option "-f a"). The data were partitioned based on the annotations imported from the reference sequence into coding (CDS), non-coding (introns), rRNA, and tRNA regions to produce a phylogeny using different estimates for each of these regions. We used the reference mitogenome of the donkey (*Equus asinus*, Genbank RefSeq: NC_001788.1) as an outgroup.

The relationships of the Puerto Real horse to the 83 other mitogenomes also was visualized using a median-joining network. This approach permits the reconstruction of the networks illustrating the relationships between organisms based on Kruskal's algorithm for generating minimum spanning trees and Farris's maximum-parsimony (MP) heuristic algorithm [49]. The network was realized using the PopART (Population Analysis with Reticulate Trees) software [50]. This visualization provides results similar to the maximum likelihood phylogenetic analysis (Fig 3).

## Results

This study instead explicitly focuses on the whole mitochondrial sequence following more recent horse phylogeographic analyses showing that the short segments of the mtDNA used in earlier studies were often accompanied by high levels of recurrent mutations, thus blurring the structure of the phylogenetic analyses and rendering some clades more difficult to distinguish from one another [23]. Thus, most complete mitogenomes published on Genbank stem from the Achilli and colleagues study mentioned above. A GenBank query using the terms "horse mitochondrion" returned a total of 511 complete mtDNA sequences (~16.6 kbp). None of the returned sequences were individuals from Iberia and all the specimens from the Americas found using this query were included in our study. The study by Achilli and colleagues [23] identified a total of 667 SNPs, including 549 sites in the coding region and 118 in the non-coding region, mostly represented by the D-loop (113 out of 960 nucleotides) and with these data identified 18 haplogroups (S1 Table). The Puerto Real horse falls into the haplogroup A (Figs 3 and 4; S2 Fig). This equine branch is defined by a specific mutation at nucleotide position 15,720. According to Achilli et al. [23], individuals of the equine haplogroup A mostly are found in Central Asia (46.7% of the modern specimens), Southern Europe (37.3%), and the Middle East (10.0%). Horses belonging to this haplogroup also were identified in Bronze Age archaeological contexts in the Iberian Peninsula [23, 51]. As with most of the other mitochondrial lineages, members of haplogroup A are characterized by a wide geographical range including, for example, the Maremmano (Italy), Akhal Teke (Turkmenistan, central Asia), Caspian ponies (Central Asia), and several breeds from the Middle East. However, this equine group does not include any Northern European breeds, indicating that the Puerto Real horse is least likely to have been bred from horses in that region. The phylogeny and haplotype maps indicate that the Puerto Real horse is most closely related to a modern specimen of the American Chincoteague pony breed (JN398377) (Fig 3), a finding backed by 86% bootstrap support (Fig 4). These two taxa differ by six mutations (three transitions at positions 6792, 16298, and 16404, and three transversions at positions 12,735, 16,343, and 16,403).

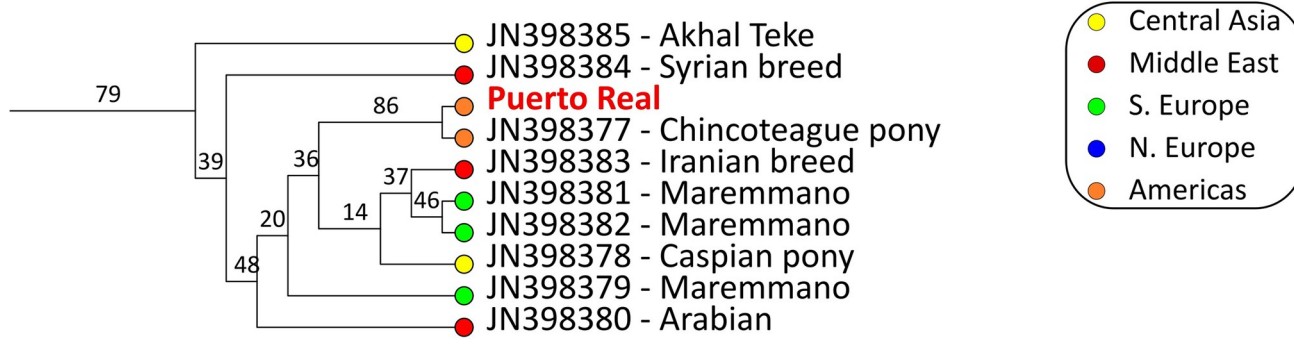

**Fig 3. Median joining network of the 85 equid specimens (including *Equus asinus* as an outgroup and the Puerto Real horse).** Numbers on the branches indicate the number of mutations separating each mitogenome.

**Fig 4. Detail of the haplogroup A phylogeny.** The complete phylogeny of all 85 sampled individuals can be found in S2 Fig.

## Discussion

The introduction of domestic horses into the Western Hemisphere was part of the massive exchange of organisms between both sides of the Atlantic Ocean that began with the European invasion of the Americas [52]. Eurasian domesticates such as cows, pigs (*Sus scrofa*), sheep (*Ovis aries*), and chicken (*Gallus gallus*) had been central to the economic, social, and symbolic life of European and African societies since the Neolithic. They were brought to the Americas by colonists whose lifestyles and economies relied on these animals. In this context, horses played a central role in colonial life, and were crucial symbols of social status for Spanish colonists. The ownership of horses often characterized high-status individuals such as *hidalgos* (members of the Iberian nobility) and access to them by Native people in the Americas was restricted [53]. Besides being symbols of elite social status, horse labor was vital in some colonial economies and conditioned colonists use of American environments. Horses also facilitated the implementation of other animal-related industries, such as ranching, that fundamentally reshaped much of the American landscape [16, 54].

Historical scholarship on the sources of Spanish colonial horses suggests they were brought directly from Castile (central Spain) via the Canary Islands. Previous DNA studies on modern breeds from the Americas also suggest a Spanish origin for early colonial horses [9, 55]. The attribution of the Puerto Real horse to the haplogroup A is consistent with an Iberian origin for horses in the Caribbean after 1492. This maternal lineage extends from Central Asia to southern Europe and has been present in the Iberian Peninsula since at least the Bronze Age.

The specimen that presents the closest affinities with the Puerto Real horse is an individual belonging to the Chincoteague pony breed. Chincoteague ponies are a feral equine breed found on the islands of Chincoteague and Assateague, off the Maryland and Virginia coasts (United States) [56]. This small population (28 individuals in 1968, 175 in 2001) has been extensively studied from a conservation and ecological perspective to assess their status as an American heritage breed [56, 57]. The origin of the Chincoteague ponies is popularized by a local oral history retold in the mid-20th century children's novel "Misty of Chincoteague" [58]. According to this story, Chincoteague ponies descend from a small herd of horses that escaped the shipwreck of a Spanish galleon during the colonial era. The galleon is said to have set sail from the Caribbean but was caught in a storm and shipwrecked close to Chincoteague Island. This narrative has generated some opposition, some authors arguing that the ponies did not precede the arrival of British settlers on the island because early British documents on the island did not describe feral horse population [59], or that it was not likely for horses to survive a shipwreck by swimming to the shore [60]. Other authors argue that the presence of a 1750 Spanish shipwreck close to the island lends veracity to the story [61, 62]. An earlier study, based on the comparative analysis of 10 protein loci revealed genetic affinities between the Chincoteague ponies and specimens of the Paso Fino breed from the Caribbean region [57].

Beyond folk stories, affinities between early Caribbean horse breeds and the Chincoteague ponies may reflect Spanish efforts to colonize the Atlantic coast of North America. These colonizing activities persisted throughout the 16th-century, reaching as far north as Chesapeake Bay. Most of these efforts originated on Hispaniola, Cuba, and Puerto Rico. Colonization, trading, and raiding among European interests in the Caribbean and along the Atlantic coast into the 1800s offered many opportunities for Spanish horses to reach the mid-Atlantic coast. By revealing these genetic affinities, this study contributes a new perspective on colonization's impact on domestication, colonial economies, and its environmental consequences.

## Conclusion

While limited to a single mitogenome, this study has a particular significance as it constitutes the first archaeogenomic evidence of post-Columbian horses in the Western Hemisphere. Our analysis suggests that the Puerto Real horse belongs to the equine haplogroup A, a maternal branch commonly found in Central Asia and Southern Europe. These results are consistent with an Iberian origin of this horse, as specimens of this haplogroup are found as early as the Bronze Age in the Iberian Peninsula. This supports early colonial accounts that suggest Spanish Andalusia was the source of the first horses brought to the Caribbean. We also found that the Puerto Real horse is closely related to another breed currently found in the Americas, the feral population of Chincoteague ponies. Further work on feral horse populations along the Atlantic coast of the United States, and continued horse archaeogenomic sampling promises to further clarify our understanding of how horses colonized the Western Hemisphere.

## Supporting information

**S1 Fig. MapDamage [37] fragment misincorporation plot for the Puerto Real horse sample.**
(TIF)

**S2 Fig. Partitioned RAxML (39) phylogeny.** Bootstrap support is on the nodes (10,000 iterations). Letters on the right correspond to the equine haplogroups defined by Achilli et al. [22].
(TIF)

**S1 Table. Samples from GenBank that were included in the phylogeographic analysis.** Except for NC001788 (*Equus asinus*) all are published in Achilli et al. [22].
(DOCX)

**S1 File.**
(DOCX)

## Acknowledgments

Permission for excavation and export of the faunal materials for curation at the Florida Museum was provided to Deagan by the government of Haiti, and permission for analysis was provided by the curators of the Historical Archaeology (Cobb) and Environmental Archaeology (Emery) programs of the Florida Museum. We gratefully acknowledge the original archaeological and zooarchaeological research conducted by Deagan as well as Charles R. Ewen, Jennifer Hamilton, Rochelle A. Marrinan, Bonnie G. McEwan, Raymond R. Willis, and Elizabeth S. Wing, and their Haitian colleagues and collaborators. We would also like to thank Florida Museum Collection Managers Gifford Waters and Nicole Cannarozzi for their assistance and valuable input to this study.

## Author Contributions

**Conceptualization:** Nicolas Delsol, Kitty F. Emery.

**Formal analysis:** Brian J. Stucky, Jessica A. Oswald.

**Funding acquisition:** Nicolas Delsol, Kitty F. Emery.

**Investigation:** Nicolas Delsol, Brian J. Stucky, Elizabeth J. Reitz.

**Methodology:** Nicolas Delsol.

**Resources:** Kitty F. Emery, Robert Guralnick.

**Supervision:** Robert Guralnick.

**Visualization:** Nicolas Delsol.

**Writing – original draft:** Nicolas Delsol.

**Writing – review & editing:** Nicolas Delsol, Brian J. Stucky, Jessica A. Oswald, Elizabeth J. Reitz, Kitty F. Emery, Robert Guralnick.

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
