## [Decision Letter · Decision Letter 0]

1 Mar 2022

PONE-D-21-38977Analysis of the earliest complete mtDNA genome of a Caribbean colonial horse (Equus caballus) from 16th century Haiti.PLOS ONE

Dear Dr. Delsol,

Thank you for submitting your manuscript to PLOS ONE. After careful consideration, we feel that it has merit but does not fully meet PLOS ONE’s publication criteria as it currently stands. Therefore, we invite you to submit a revised version of the manuscript that addresses the points raised during the review process. Please submit your revised manuscript by Apr 15 2022 11:59PM. If you will need more time than this to complete your revisions, please reply to this message or contact the journal office at plosone@plos.org. Please include the following items when submitting your revised manuscript:A rebuttal letter that responds to each point raised by the academic editor and reviewer(s). You should upload this letter as a separate file labeled 'Response to Reviewers'.A marked-up copy of your manuscript that highlights changes made to the original version. You should upload this as a separate file labeled 'Revised Manuscript with Track Changes'.An unmarked version of your revised paper without tracked changes. You should upload this as a separate file labeled 'Manuscript'.

We look forward to receiving your revised manuscript.

Kind regards,

Tzen-Yuh Chiang

Academic Editor

PLOS ONE

Journal Requirements:

2. In your manuscript, please provide additional information regarding the specimens used in your study. Ensure that you have reported specimen numbers and complete repository information, including museum name and geographic location.

For more information on PLOS ONE's requirements for paleontology and archaeology research, see https://journals.plos.org/plosone/s/submission-guidelines#loc-paleontology-and-archaeology-research.

3. Please include a complete copy of PLOS’ questionnaire on inclusiveness in global research in your revised manuscript. Our policy for research in this area aims to improve transparency in the reporting of research performed outside of researchers’ own country or community. The policy applies to researchers who have traveled to a different country to conduct research, research with Indigenous populations or their lands, and research on cultural artifacts. The questionnaire can also be requested at the journal’s discretion for any other submissions, even if these conditions are not met.  Please find more information on the policy and a link to download a blank copy of the questionnaire here: https://journals.plos.org/plosone/s/best-practices-in-research-reporting. Please upload a completed version of your questionnaire as Supporting Information when you resubmit your manuscript.

   1. You may seek permission from the original copyright holder of Figures 1 to publish the content specifically under the CC BY 4.0 license.  

Maps at the CIA (public domain): https://www.cia.gov/library/publications/the-world- factbook/index.html and https://www.cia.gov/library/publications/cia-maps-publications/index.html

“This research was funded by the National Science Foundation to Emery and Delsol (DDIG 1930628), the Fulbright program (to Delsol), and additional support from the Florida Museum Ancient DNA Laboratory (from Guralnick). Permission for excavation and export of the faunal materials for curation at the Florida Museum was provided to Deagan by the government of Haiti, and permission for analysis was provided by the curators of the Historical Archaeology (Cobb) and Environmental Archaeology (Emery) programs of the Florida Museum.”

“KFE and ND received support from NSF DDRIG grant (#1930628) for this research (https://beta.nsf.gov/funding/opportunities/archaeology-and-archaeometry-0). The funders had no role in study design, data collection and analysis, decision to publish, or preparation of the manuscript.”

Reviewers' comments:

Reviewer's Responses to Questions

**Comments to the Author**

1. Is the manuscript technically sound, and do the data support the conclusions?

Reviewer #1: Yes

Reviewer #2: Yes

Reviewer #3: Partly

Reviewer #4: Yes

2. Has the statistical analysis been performed appropriately and rigorously? 

Reviewer #1: Yes

Reviewer #2: Yes

Reviewer #3: N/A

Reviewer #4: Yes

3. Have the authors made all data underlying the findings in their manuscript fully available?

Reviewer #1: Yes

Reviewer #2: No

Reviewer #3: No

Reviewer #4: Yes

4. Is the manuscript presented in an intelligible fashion and written in standard English?

Reviewer #1: Yes

Reviewer #2: Yes

Reviewer #3: Yes

Reviewer #4: Yes

5. Review Comments to the Author

Reviewer #1: It is an interesting research. Author briefed an interesting history about horses in the Americas and take us an overview about the origin of early translocated individual and its phylogenetic relationship with modern breeds in the Americas.

Reviewer #2: The manuscript reads very well and presents a nice story that fits with the historical records for the introduction of the horse into the Americas in the 16th century. The only minor issue with the results presented is that it is only based on a single specimen.

While the authors indicate that the sequence is publicly available at the NCBI, no accession number is provided. The accession number needs to be added before the paper can be published.

For the rest, I only have a few minor comments:

Line 73: "By the 1650s century" delete "century"

Lines 167-175. It would have been nice if the authors would have added a map of the archeological site in the supplementary material.

Lines 232-237: I assume the medium-joining network mentioned in this paragraph is the one shown in figure 3?

Line 259: "Figure S1" should be "Figure S2"

Reviewer #3: The paper reports assembly of the whole mt genome of a single horse dating back to 16th century. It is well written, and although only one sample was analyzed I think it adds to current knowledge.

However, I have several suggestions to improve the paper before publication on PLOS ONE, mainly related to reproducibility and data availability.

In particular, authors declared that they made the data publicly available, but I wasn't able to find any link pointing to them.

I list my concerns below regarding Data availability and Methods (and reproducibility)

Data availability issues:

Raw reads should be deposited on SRA. The assembled genome should be deposited in GenBank. Authors declared that the data are available, but they are not.

Methods and reproducibility issues:

First of all, I think authors performed all the analysis (mapping, assembling, phylogenetic tree construction) using the Geneious commercial software. This should be clearly declared.

Most importantly, for each analysis step, all the parameters and options used should be clarified so that a user buying Genious and having access to the data would be able to repeat the experiment. I think we are still far from this. I give some suggestions below, but I invite the authors to carefully check the methods and identify more missing details.

Line 210: “The 3’ and 5’ reads were imported, paired, and deduplicated in Geneious (https://www.geneious.com/).” This description should be more detailed. 1) What does it mean that read were “imported, paired”? Reads are already provided as paired by the sequencing instrument. Did the authors mean that the tried to merge paired reads in a single read? 2) What does it mean that reads were “deduplicated”? Did authors remove duplicates? 3) I assume 3’ and 5’ reads are the so-called forward and reverse reads. I think it is more common to name them forward and reverse, but this is not an issue.

Line 211: “The sequence was attributed to a horse after a BLAST search on the reads.” Did the authors use the blast online service? If so, they should provide the link and/or a reference to BLAST publications. Did the authors use default parameters? Did the authors blast against nt or specifically against the horse genome? Which threshold did they set in order to accept the hits as horse?

Lines 211-213: Which software was used for mapping? Which parameters were used to ensure that the maximum proportion of mismatches was 5%?

Lines 213-214: It is not clear how authors went from mapping reads to assemble the mitochondrial genome. This should be described.

Lines 216-218: Which software used for building the consensus? How were contigs generated? Which requisites were required for a contig to be used in the assembly? (e.g. contig length, contig coverage). Which overlap between contigs was required to assemble them into a scaffold? “We built the consensus sequence of the whole mitochondrion of the Puerto Real horse with a threshold of 75% and a quality set on “Highest”. The consensus was generated out of contig pileups that represented a mean coverage of 5622.9X of the horse reference mitochondrion.”

Lines 229-230: “The mitochondrial data was partitioned into coding, non-coding, and RNA regions to produce a phylogeny using different model estimates for each of these regions.” What are the differences between the model estimates? Are they implemented in Geneious? What are their names?

Reviewer #4: This study focused on a single ancient horse sample from 16th century /Caribbean colonial horse/. Complete mitochondrial sequencing of this single sample was performed. According to the authors, this mitochondrial genotype is associated with group A, and is the most closely related to a modern specimen of the American Chincoteague pony breed from North America (JN398377). I did not find a deposit of your genomic sequence and Genbank. Why?

The article is well discussed historically, but in contrast, relatively genetic research in this region is lacking. It would be good to make a comparative analysis of the available genetic data on the biodiversity of mitochondrial lines (modern and ancient) for America and the Iberian Peninsula. I would recommend adding data on modern local horses from the study region.

6. PLOS authors have the option to publish the peer review history of their article (what does this mean?). If published, this will include your full peer review and any attached files.

Reviewer #1: No

Reviewer #2: No

Reviewer #3: No

Reviewer #4: No

---

## [Author Response · Author response to Decision Letter 0]

11 Apr 2022

We would like to thank the academic editor and the reviewers for their thoughtful comments which have greatly helped improve our manuscript. Please find here our response to the points they raised. For clarity purposes, the editor and reviewers’ initial comments are inserted before our responses.

A. Editor’s comment:

We made sure to implement PLOS ONE’s style template in the main manuscript and changed the figures file names to follow the file naming template.

2. In your manuscript, please provide additional information regarding the specimens used in your study. Ensure that you have reported specimen numbers and complete repository information, including museum name and geographic location.

For more information on PLOS ONE's requirements for paleontology and archaeology research, see https://journals.plos.org/plosone/s/submission-guidelines#loc-paleontology-and-archaeology-research.

We added a Methods subsection that details additional information on the permits and curation institution, including the statement 'All necessary permits were obtained for the described study, which complied with all relevant regulations.'

3. Please include a complete copy of PLOS’ questionnaire on inclusiveness in global research in your revised manuscript. Our policy for research in this area aims to improve transparency in the reporting of research performed outside of researchers’ own country or community. The policy applies to researchers who have traveled to a different country to conduct research, research with Indigenous populations or their lands, and research on cultural artifacts. The questionnaire can also be requested at the journal’s discretion for any other submissions, even if these conditions are not met. Please find more information on the policy and a link to download a blank copy of the questionnaire here: https://journals.plos.org/plosone/s/best-practices-in-research-reporting. Please upload a completed version of your questionnaire as Supporting Information when you resubmit your manuscript.

The questionnaire on inclusiveness in global research will be uploaded in the SI section together with the revised documents.

 1. You may seek permission from the original copyright holder of Figures 1 to publish the content specifically under the CC BY 4.0 license. 

Maps at the CIA (public domain): https://www.cia.gov/library/publications/the-world- factbook/index.html and https://www.cia.gov/library/publications/cia-maps-publications/index.html

Figure 1 is a map that was designed by the lead author (Nicolas Delsol) using the open-source GIS software QGIS (version 3.24.1). The rasters used for the design of this map come from public and non-copyright cartographic resources generated by the NASA and and Japan’s Ministry of Economy, Trade, and Industry (METI). These resources were retrieved at the url http://eros.usgs.gov/# and were recommended by the Academic Editor in his comments. In the caption of the figure we mention the source of this raw cartographic data.

“This research was funded by the National Science Foundation to Emery and Delsol (DDIG 1930628), the Fulbright program (to Delsol), and additional support from the Florida Museum Ancient DNA Laboratory (from Guralnick). Permission for excavation and export of the faunal materials for curation at the Florida Museum was provided to Deagan by the government of Haiti, and permission for analysis was provided by the curators of the Historical Archaeology (Cobb) and Environmental Archaeology (Emery) programs of the Florida Museum.”

“KFE and ND received support from NSF DDRIG grant (#1930628) for this research (https://beta.nsf.gov/funding/opportunities/archaeology-and-archaeometry-0). The funders had no role in study design, data collection and analysis, decision to publish, or preparation of the manuscript.”

The Acknowledgement section has been changed to reflect the Editor’s requests. We also removed funding-related text from the manuscript.

In addition, we ask the Academic Editor to change the funding statement in the following way:

“KFE and ND received support from NSF DDRIG grant (#1930628) for this research (https://beta.nsf.gov/funding/opportunities/archaeology-and-archaeometry-0). ND’s initial research was also funded by a Fulbright scholarship. The field research in Haiti was financed through contributions of the Organization of American States, the National Endowment for the Humanities, Dr. William Goya, the Wentworth Foundation, the University of Florida Division of Sponsored Research, and the Florida Museum of Natural History. The funders had no role in study design, data collection and analysis, decision to publish, or preparation of the manuscript.”

B. Reviewers’ comments:

- Reviewers #2, #3, and #4 expressed some concern about the availability of the consensus sequence and the raw reads on public databases. The raw reads have been deposited on the NCBI SRA repository (BioProject# PRJNA817535, SRA# SRR18395639). The annotated mitochondrial consensus sequence has been deposited on Genbank under the reference # ON168403. Both datasets are set to be publicly released upon publication of the article.

- Reviewer #2: 

The manuscript reads very well and presents a nice story that fits with the historical records for the introduction of the horse into the Americas in the 16th century. The only minor issue with the results presented is that it is only based on a single specimen.

While the authors indicate that the sequence is publicly available at the NCBI, no accession number is provided. The accession number needs to be added before the paper can be published.

For the rest, I only have a few minor comments:

Line 73: "By the 1650s century" delete "century"

Lines 167-175. It would have been nice if the authors would have added a map of the archeological site in the supplementary material.

Lines 232-237: I assume the medium-joining network mentioned in this paragraph is the one shown in figure 3?

Line 259: "Figure S1" should be "Figure S2"

We made the minor revisions asked for on lines 73 and 259. We also added a reference to Fig 3 to refer to the median-joining network. About the site map, we did not have access to a high-quality reproduction of a map of the site, given that most publications on this site are almost 30 years old. The syntheses of the site that were cited (Deagan 1995, 1996) provide some cartographic data.

- Reviewer #3: 

The paper reports assembly of the whole mt genome of a single horse dating back to 16th century. It is well written, and although only one sample was analyzed I think it adds to current knowledge.

However, I have several suggestions to improve the paper before publication on PLOS ONE, mainly related to reproducibility and data availability.

In particular, authors declared that they made the data publicly available, but I wasn't able to find any link pointing to them.

I list my concerns below regarding Data availability and Methods (and reproducibility)

Data availability issues:

Raw reads should be deposited on SRA. The assembled genome should be deposited in GenBank. Authors declared that the data are available, but they are not.

Methods and reproducibility issues:

First of all, I think authors performed all the analysis (mapping, assembling, phylogenetic tree construction) using the Geneious commercial software. This should be clearly declared.

Most importantly, for each analysis step, all the parameters and options used should be clarified so that a user buying Genious and having access to the data would be able to repeat the experiment. I think we are still far from this. I give some suggestions below, but I invite the authors to carefully check the methods and identify more missing details.

Line 210: “The 3’ and 5’ reads were imported, paired, and deduplicated in Geneious (https://www.geneious.com/).” This description should be more detailed. 1) What does it mean that read were “imported, paired”? Reads are already provided as paired by the sequencing instrument. Did the authors mean that the tried to merge paired reads in a single read? 2) What does it mean that reads were “deduplicated”? Did authors remove duplicates? 3) I assume 3’ and 5’ reads are the so-called forward and reverse reads. I think it is more common to name them forward and reverse, but this is not an issue.

Line 211: “The sequence was attributed to a horse after a BLAST search on the reads.” Did the authors use the blast online service? If so, they should provide the link and/or a reference to BLAST publications. Did the authors use default parameters? Did the authors blast against nt or specifically against the horse genome? Which threshold did they set in order to accept the hits as horse?

Lines 211-213: Which software was used for mapping? Which parameters were used to ensure that the maximum proportion of mismatches was 5%?

Lines 213-214: It is not clear how authors went from mapping reads to assemble the mitochondrial genome. This should be described.

Lines 216-218: Which software used for building the consensus? How were contigs generated? Which requisites were required for a contig to be used in the assembly? (e.g. contig length, contig coverage). Which overlap between contigs was required to assemble them into a scaffold? “We built the consensus sequence of the whole mitochondrion of the Puerto Real horse with a threshold of 75% and a quality set on “Highest”. The consensus was generated out of contig pileups that represented a mean coverage of 5622.9X of the horse reference mitochondrion.”

Lines 229-230: “The mitochondrial data was partitioned into coding, non-coding, and RNA regions to produce a phylogeny using different model estimates for each of these regions.” What are the differences between the model estimates? Are they implemented in Geneious? What are their names?

Reviewer #3 had several recommendations to improve the paper, particularly in terms of reproducibility and data availability. We already addressed the question of data accessibility (see above). In the methods section, we added some clarifications as requested by the reviewer as shown in the new version of the manuscript. More specifically, we added additional details on how the reads were assembled using the commercial software Geneious (lines 210-218).

- Reviewer #4: 

This study focused on a single ancient horse sample from 16th century /Caribbean colonial horse/. Complete mitochondrial sequencing of this single sample was performed. According to the authors, this mitochondrial genotype is associated with group A, and is the most closely related to a modern specimen of the American Chincoteague pony breed from North America (JN398377). I did not find a deposit of your genomic sequence and Genbank. Why?

The article is well discussed historically, but in contrast, relatively genetic research in this region is lacking. It would be good to make a comparative analysis of the available genetic data on the biodiversity of mitochondrial lines (modern and ancient) for America and the Iberian Peninsula. I would recommend adding data on modern local horses from the study region.

Reviewer #4 also expressed some concern about the accessibility of the sequence and the reads (see above for the response). We agree with their suggestion that adding more mitochondrial genomes of horses from the Caribbean would be helpful. However, there are few published and available mitogenomes from the Caribbean on public repositories such as Genbank or the ENA, and we used all available, relevant mitogenomes in our analysis. We hope that further work on horse heritage breeds from across the Caribbean will provide more data to fill this gap.

Again, we thank the academic editor and the reviewers for their comments and suggestions. We sincerely hope the changes in this revised version of the manuscript reflects these recommendations.

Sincerely,

---

## [Decision Letter · Decision Letter 1]

17 May 2022

PONE-D-21-38977R1Analysis of the earliest complete mtDNA genome of a Caribbean colonial horse (Equus caballus) from 16th-century Haiti.PLOS ONE

Dear Dr. Delsol,

Thank you for submitting your manuscript to PLOS ONE. After careful consideration, we feel that it has merit but does not fully meet PLOS ONE’s publication criteria as it currently stands. Therefore, we invite you to submit a revised version of the manuscript that addresses the points raised during the review process.

We look forward to receiving your revised manuscript.

Kind regards,

Tzen-Yuh Chiang

Academic Editor

PLOS ONE

Reviewers' comments:

Reviewer's Responses to Questions

**Comments to the Author**

1. If the authors have adequately addressed your comments raised in a previous round of review and you feel that this manuscript is now acceptable for publication, you may indicate that here to bypass the “Comments to the Author” section, enter your conflict of interest statement in the “Confidential to Editor” section, and submit your "Accept" recommendation.

Reviewer #1: All comments have been addressed

Reviewer #3: All comments have been addressed

Reviewer #4: (No Response)

2. Is the manuscript technically sound, and do the data support the conclusions?

Reviewer #1: Yes

Reviewer #3: Yes

Reviewer #4: Partly

3. Has the statistical analysis been performed appropriately and rigorously? 

Reviewer #1: Yes

Reviewer #3: N/A

Reviewer #4: No

4. Have the authors made all data underlying the findings in their manuscript fully available?

Reviewer #1: Yes

Reviewer #3: Yes

Reviewer #4: No

5. Is the manuscript presented in an intelligible fashion and written in standard English?

Reviewer #1: Yes

Reviewer #3: Yes

Reviewer #4: Yes

6. Review Comments to the Author

Reviewer #1: (No Response)

Reviewer #3: All my comments and concerns have been addressed, and I think that the paper is now suitable for publication. Authors provided the required details for reproducing the work using Geneious and the other software they used. Authors also provided accession numbers for the raw reads and the genome assembly. The data will be made available after publication.

Reviewer #4: The authors of the article did not comment on my two remarks.

- I did not find a deposit of your genomic sequence and Genbank?

- It would be good to make a comparative analysis of the available

genetic data on the biodiversity of mitochondrial lines (modern and ancient) for

America and the Iberian Peninsula. I would recommend adding data on modern local

horses from the study region.

In addition, the article is not written according to the requirements of PlosOne. As an example, too short a description of the results and the discussion. The abstract section is not formatted according to the requirements of the journal.

The Materials and Methods section describes the isolation of ancient DNA that does not conform to the protocols of good practice for working with ancient DNA. Тhere are no control reactions for DNA contamination as example.

I cannot believe that DNA isolation is possible in the described way to dissolve 500 mg of dry bone material in 1 ml of buffer.

7. PLOS authors have the option to publish the peer review history of their article (what does this mean?). If published, this will include your full peer review and any attached files.

Reviewer #1: No

Reviewer #3: No

Reviewer #4: No

---

## [Author Response · Author response to Decision Letter 1]

6 Jun 2022

We would like to renew our thanks to the academic editor and the reviewers for their thoughtful comments which have greatly helped improve our manuscript. We were happy to see that two of the three reviewers found our prior submission satisfactory. Please find here our responses to the points raised by Reviewer #4 in this version.

I did not find a deposit of your genomic sequence and Genbank?

 As stated in the letter attached to our first resubmission, the assembly of the horse mitogenome has been submitted successfully to Genbank (accession number ON168403) and the raw reads were submitted to the Sequence Read Archive (accession number PRJNA817535). These have been explicitly called out in the latest revisions to this article (lines 245-247). As customary, we have embargoed the release of these data until after the publication of the manuscript. As a proof of these submissions, we provide the confirmation emails we received from both repositories. 

It would be good to make a comparative analysis of the available genetic data on the biodiversity of mitochondrial lines (modern and ancient) for America and the Iberian Peninsula. I would recommend adding data on modern local horses from the study region.

 We agree with Reviewer 4 that such a comparative analysis of this 16th-century mitogenome with modern and ancient mtDNA from archaeological and modern horses from Iberia and the Americas would be extremely interesting. In the introduction section, we mention an older study that outlined some genetic similarities between American and Iberian horses using short regions of the D-loop between 360- to 442-bp long (Luís C, Bastos-Silveira C, Cothran EG, Oom M do M. Iberian Origins of New World Horse Breeds. J Hered. 2006 Mar 1;97(2):107–13).

 To address Reviewer 4’s concerns, we have further clarified our approach and have explicitly explained the value of using complete mtDNA only (lines 126-128 of the revised manuscript). We also added the following paragraph at the beginning of the Results section (lines 268-275).

“This study instead explicitly focuses on the whole mitochondrial sequence following more recent horse phylogeographic analyses showing that the short segments of the mtDNA used in earlier studies were often accompanied by high levels of recurrent mutations, thus blurring the structure of the phylogenetic analyses and rendering some clades more difficult to distinguish from one another (23). Thus, most complete mitogenomes published on Genbank stem from the Achilli and colleagues study mentioned above. A GenBank query using the terms “horse mitochondrion” returned a total of 511 complete mtDNA sequences (~16.6 kbp). None of the returned sequences were individuals from Iberia and all the specimens from the Americas found using this query were included in our study.”

In addition, the article is not written according to the requirements of PlosOne. As an example, too short a description of the results and the discussion. The abstract section is not formatted according to the requirements of the journal.

Thank you for pointing out the formatting requirements for this journal. In this revision we have made every effort to ensure our formatting falls within the guidelines found on https://journals.plos.org/plosone/s/submission-guidelines. We have also refined the text in several places. We have expanded the results section, but feel that the discussion provides sufficient content to contextualize our findings.

The Materials and Methods section describes the isolation of ancient DNA that does not conform to the protocols of good practice for working with ancient DNA. 

Тhere are no control reactions for DNA contamination as example.

I cannot believe that DNA isolation is possible in the described way to dissolve 500 mg of dry bone material in 1 ml of buffer.

 Our methods for ancient DNA work have been outlined in a number of publications (Oswald et al, 2019, 2020, 2021). We have further described the Florida Museum Ancient DNA Laboratory and its protocols. As stated in our earlier MS and clarified in this version, this study did include negative controls. As per the Florida Museum aDNA laboratory, all extractions (for this study and others) include negative controls. As shown by our results in these several studies, our ratio of sample to buffer is appropriate and effective. The protocols used for DNA extraction are based not only on our work but also on published studies such as those of Yang et al. 1998, 2008 and others. 

1. Yang, D. Y., Eng, B., Waye, J. S., Dudar, J. C. & Saunders, S. R. Technical note: Improved DNA extraction from ancient bones using silica-based spin columns. Am. J. Phys. Anthropol. 105, 539–43 (1998).

2. Yang, D. Y., Liu, L., Chen, X. & Speller, C. F. Wild or domesticated: DNA analysis of ancient water buffalo remains from north China. J. Archaeol. Sci. 35, 2778–2785 (2008).

To clarify our protocols and further describe the measures we used to limit cross-contamination, we have added details about our negative control reactions during the extraction process (lines 205-207).

Again, we thank the academic editor and the reviewers for their comments and suggestions. We hope the changes we have made in this revised version of the manuscript effectively addresses these recommendations.

Sincerely,

Nicolas Delsol

Corresponding author

---

## [Decision Letter · Decision Letter 2]

14 Jun 2022

Analysis of the earliest complete mtDNA genome of a Caribbean colonial horse (Equus caballus) from 16th-century Haiti.

PONE-D-21-38977R2

Dear Dr. Delsol,

We’re pleased to inform you that your manuscript has been judged scientifically suitable for publication and will be formally accepted for publication once it meets all outstanding technical requirements.

Kind regards,

Tzen-Yuh Chiang

Academic Editor

PLOS ONE

Additional Editor Comments (optional):

Reviewers' comments:

Reviewer's Responses to Questions

**Comments to the Author**

1. If the authors have adequately addressed your comments raised in a previous round of review and you feel that this manuscript is now acceptable for publication, you may indicate that here to bypass the “Comments to the Author” section, enter your conflict of interest statement in the “Confidential to Editor” section, and submit your "Accept" recommendation.

Reviewer #3: All comments have been addressed

2. Is the manuscript technically sound, and do the data support the conclusions?

Reviewer #3: Yes

3. Has the statistical analysis been performed appropriately and rigorously? 

Reviewer #3: Yes

4. Have the authors made all data underlying the findings in their manuscript fully available?

Reviewer #3: Yes

5. Is the manuscript presented in an intelligible fashion and written in standard English?

Reviewer #3: Yes

6. Review Comments to the Author

Reviewer #3: (No Response)

7. PLOS authors have the option to publish the peer review history of their article (what does this mean?). If published, this will include your full peer review and any attached files.

Reviewer #3: No

---

## [Editor Report · Acceptance letter]

30 Jun 2022

PONE-D-21-38977R2 

Analysis of the earliest complete mtDNA genome of a Caribbean colonial horse (*Equus caballus*) from 16^th^-century Haiti 

Dear Dr. Delsol:

I'm pleased to inform you that your manuscript has been deemed suitable for publication in PLOS ONE. Congratulations! Your manuscript is now with our production department. 

Kind regards, 

on behalf of

Dr. Tzen-Yuh Chiang 

Academic Editor

PLOS ONE